# Revolutionising Cardio-Oncology Care with Precision Genomics

**DOI:** 10.3390/ijms26052052

**Published:** 2025-02-26

**Authors:** Jun Hua Chong, Charles T. H. Chuah, Caroline G. Lee

**Affiliations:** 1National Heart Centre Singapore, 5 Hospital Dr, Singapore 169609, Singapore; 2Duke-NUS Medical School, 8 College Road, Singapore 169857, Singapore; 3National Cancer Centre Singapore, 30 Hospital Blvd, Singapore 168583, Singapore; 4Singapore General Hospital, Outram Road, Singapore 169608, Singapore; 5Department of Biochemistry, Yong Loo Lin School of Medicine, National University of Singapore, C/O MD7, Level 2, 8 Medical Drive, Singapore 117597, Singapore; 6Division of Cellular & Molecular Research, Humphrey Oei Institute of Cancer Research, National Cancer Centre Singapore, 11 Hospital Crescent, Singapore 169610, Singapore; 7NUS Graduate School, National University of Singapore, 21 Lower Kent Ridge Road, Singapore 119077, Singapore

**Keywords:** precision genomics, cardio-oncology, cancer-therapy-related cardiac dysfunction, clonal haematopoiesis of indeterminate potential, polygenic risk scoring, long noncoding RNAs and microRNAs, human-induced pluripotent stem cells

## Abstract

Cardiovascular disease is the worldwide leading cause of mortality among survivors of cancer due in part to the cardiotoxicity of anticancer therapies. This paper explores the progress in precision cardio-oncology, particularly in genetic testing and therapeutics, and its impact on cardiovascular diseases in clinical and laboratory settings. These advancements enable clinicians to better assess risk, diagnose conditions, and deliver personalised, cost-effective therapeutics. Through case studies of cancer-therapy-related cardiac dysfunction, clonal haematopoiesis of indeterminate potential, and polygenic risk scoring, we demonstrate the benefits of incorporating precision genomics in individualised care in cardio-oncology. Furthermore, leveraging real-world genomic data in clinical settings can advance our understanding of long noncoding RNAs and microRNAs, which play important regulatory roles in cardio-oncology. Additionally, employing human-induced pluripotent stem cells to stratify risk and guide prevention strategies represents a promising avenue for modelling precision cardio-oncology. While these advancements showcase the significant progress in genetic approaches, they also raise substantial ethical, legal, and societal concerns. Regulatory oversight of genetic and genomic technologies should therefore evolve suitably to keep up with rapid advancements in technology and analysis. Provider education is crucial for the appropriate use of new genetic and genomic applications, including on the existing protection available for patients regarding genetic information. This can provide confidence for diverse study groups to advance genetic studies looking to develop a comprehensive understanding and effective clinical applications for heterogeneous populations. In clinical settings, the implementation of genetic and genomic applications within electronic medical records can offer point-of-care clinical decision support, thus providing timely information to guide clinical management decisions.

## 1. Introduction and Background

Cardiovascular disease is the main cause of death among survivors of cancer, accounting internationally for over 370,000 deaths per year, with new cancer formation risk being the second major concern in this patient group [1]. This elevated risk of cardiovascular disease is partly attributed to the prior use of anticancer therapies, as nearly all cancer therapies, including chemotherapies, immune checkpoint inhibitors, and monoclonal antibodies, are associated with cardiovascular complications [2]. The growing recognition of this issue has stimulated the emergence of the field of cardio-oncology, which focuses on understanding the molecular mechanisms underpinning the cardiac dysfunction induced by anticancer treatments. Cardio-oncology also seeks to develop innovative strategies to mitigate the cardiotoxic effects of these therapies.

According to the inaugural European Society of Cardiology Guidelines in Cardio-Oncology 2022 [3], genetic assessment should be included in baseline cardiovascular toxicity risk evaluations. Table 1 highlights the key genetic considerations in baseline cardiovascular toxicity risk assessment before cancer therapy initiation. A family history of premature cardiovascular disease (CVD) is an important consideration, as genetic abnormalities may increase the risk of cancer-therapy-related cardiovascular complications [4,5,6]. The incidence of ventricular arrhythmias during cancer treatment increases in patients with advanced cancer and existing cardiovascular comorbidities [7,8,9,10]. When a baseline QT interval corrected using Fridericia’s formula (QTcF) prolongation (a type of ECG measurement indicating the potential for ventricular arrhythmias) is identified, it is advisable to address any reversible causes and identify the genetic conditions that prolong QTcF [11]. Thromboembolic events during cancer treatment, known as cancer-associated thrombosis (CAT), are driven by a complex interplay of factors, including the cancer-induced prothrombotic state, the propensity of specific anticancer treatments to result in thrombus formation, and patient-specific risks such as demographics, genetic predisposition, and concurrent medical conditions [12]. Genetic predisposition to atrial fibrillation (AF) contributes to the pathophysiology of AF associated with cancer, and clinicians should have higher suspicion regarding the development of AF in people with cancer with an associated genetic predisposition. In addition to transient ECG monitoring (12-lead) or prolonged ECG monitoring (Holter) to assess the QT interval and the development of AF, regular blood pressure monitoring to stratify cardiovascular risk and echocardiographic assessment of the left ventricular function to monitor for heart failure development should be considered.

Ongoing research into the mechanisms linking cardiovascular disease and cancer, particularly the genetic predisposition to cardiovascular toxicity, is required to further our understanding of this area. Investigating the role of genetics in cardiovascular toxicity is essential for enhancing risk stratification and developing genetic profiles that enable the early detection of this toxicity. Integrating genetic profiling into the prediction of cancer-therapy-related cardiac dysfunction (CTRCD) is imperative, as it has the potential to surpass the predictive accuracy of traditional cardiovascular risk factors.

Precision medicine seeks to improve the accuracy and specificity of disease characterisation by integrating diverse data sources, including genomics and digital health metrics. This approach enhances the precision and accuracy of the diagnoses, definitions, and treatment of various disease subtypes. By understanding the molecular foundations of a patient’s condition, treatments can be tailored to individual needs, moving away from broad, one-size-fits-all solutions. Precision methods in cardio-oncology thus help to address the prediction/prevention, early detection, monitoring, and treatment of CTRCD [13].

This paper explores the progress in precision cardio-oncology, particularly in genetic testing and therapeutics, and its impact on both common and uncommon cardiovascular conditions in clinical and laboratory-based settings. These advancements enable clinicians to better assess risk, diagnose conditions, and deliver personalised therapeutics. Through case studies of CTRCD, clonal haematopoiesis of indeterminate potential (CHIP), and polygenic risk scoring (PRS), we demonstrate the benefits of incorporating precision genomics in individualised care in cardio-oncology.

Furthermore, leveraging real-world genomic data in clinical settings can advance our understanding of long noncoding RNAs and microRNAs, which play important regulatory roles in cardio-oncology. Additionally, employing human-induced pluripotent stem cells (hiPSCs) for stratifying risk and guiding prevention strategies represents a promising avenue for modelling precision cardio-oncology [14]. By using hiPSCs to create personalised models, it can become possible to assess the cardiovascular toxicity of treatments tailored to individual patients, potentially leading to more precise treatment and prevention strategies as well as reducing CVD morbidity and mortality [14]. The concurrent development of these emerging research fields will provide timely translational clinical benefit in this rapidly evolving area.

## 2. Rationale and Need for Precision Genomics in Cardio-Oncology Care

Three use cases are presented to highlight the benefits of leveraging precision genomics to enhance individualised clinical cardio-oncology care.

### 2.1. Use Case 1: Cancer-Therapy-Related Cardiac Dysfunction

Anthracycline-induced cardiotoxicity is a significant concern in cancer therapy, manifesting as either acute toxicity, resulting in cardiac arrhythmias or reduced left ventricular ejection fraction (LVEF), or chronic toxicity due to excess cumulative cancer therapy exposure. The risk of heart failure increases with cumulative doses, rising from 5% at 400 mg/m^2^ to 25% at 700 mg/m^2^ [15]. However, only 50% of the patients receiving increased cumulative doses suffer cardiac toxicity, suggesting that genetic factors may influence drug exposure, efficacy, and response [16].

Extensive research has been conducted on the germline predictors of anthracycline-induced cardiotoxicity [17,18,19,20,21,22,23]. Several single-nucleotide polymorphisms (SNPs), mainly implicated in drug metabolism and the ATP transport pathway, have been shown to be significantly correlated to anthracycline-induced toxicity across multiple groups [24]. A children’s oncology genome-wide association study (GWAS) identified a CELF4 (CUGBP Elav-like family member 4) polymorphism that influences the cumulative dose-related association between anthracycline exposure and cardiotoxicity. Specifically, the rs1786814 GG genotype led to a 10.2-times increase in the risk of cardiotoxicity at doses exceeding 300 mg/m^2^ [23].

Another notable GWAS identified that a non-synonymous coding variant in retinoic acid receptor gamma (RARG), rs2229774, was associated with anthracycline-induced cardiotoxicity in a paediatric cancer cohort. The SNP demonstrated a 7.0-times increase in the Canadian paediatric discovery group, and 4.1- and 5.4-times increase in Dutch and Hispanic–American replication groups [25]. As a mechanism of cardiomyopathy, the rs2229774 variant was demonstrated to cause the downregulation of DNA topoisomerase-2 beta expression in cardiomyoblasts [25].

With the advent of immunotherapies such as immune checkpoint inhibitors (ICIs), it is important to note that genetic predispositions towards CTRCD may not solely lie in germline genes. Selective clonal T-cell populations infiltrating the myocardium in patients with ICI-associated myocarditis were found to be similar to the T-cell populations in tumours and skeletal muscle. RNA sequencing studies identified the dysregulated expression of cardiac-specific genes within the tumour [26], which could have contributed to CTRCD.

The genetic variants associated with cardiovascular disease during cancer therapy have recently been reviewed [6]. Figure 1 illustrates the key genetic variants and associated cardiovascular complications. Although routine genetic testing to assess the risk of cancer-therapy-related cardiovascular toxicity before treatment is not currently recommended, striking a balance between preventing cardiotoxicity and managing cancer effectively remains essential. In patients with a genetic predisposition, a standard dose of cancer therapy may induce cardiotoxicity, while a reduced dose may compromise cancer outcomes. Therefore, a personalised genetic approach can help determine individual susceptibility to cardiovascular disease in people with cancer, thus allowing clinicians to potentially balance the efficacy and toxicity of cancer drugs. The efficacy of this approach in optimising both cancer and cardiac outcomes requires formal study in clinical trials.

### 2.2. Use Case 2: Clonal Haematopoiesis of Indeterminate Potential

CHIP is an ageing-associated entity, which leads to the development of a subpopulation of blood cells that are genetically distinct. It can occur in individuals with no haematological conditions and is recognised as a risk factor for various diseases associated with ageing, including haematological cancers and atherosclerotic cardiovascular disease (ASCVD) (coronary heart disease and stroke) [27,28,29,30]. In 2015, a formal definition of CHIP was proposed with the following criteria: (1) CHIP occurs without morphological variation in haematological cells; (2) there must be a candidate driver gene mutation with a variant allele frequency (VAF) of at least 2% in the peripheral blood; (3) the absence of the diagnostic criteria for haematologic malignancy must be confirmed [31]. While CHIP is associated with an elevated relative risk of incident haematologic cancer (HR~13) [28,29], the absolute risk elevated is modest by comparison —approximately 0.5–1% per annum [32]. Initial epidemiology reported a 40% increase in mortality among individuals with CHIP, a figure that cannot be attributed solely to the increased risk of haematologic malignancy. In these groups, only 1 out of 246 individuals with CHIP-associated mutations passed away from haematologic malignancy [28].

The ensuing studies have established a significant link between CHIP and an elevated risk of CVD, including coronary artery disease (CAD) (HR 1.8–2.0), ischaemic stroke (HR 2.6), and premature myocardial infarction (HR 4.0), independent of traditional cardiovascular risk factors [27,28,32]. The cardiovascular risk associated with CHIP is comparable to traditional Framingham risk factors [28]. Additionally, CHIP carrier status has been associated with a worse prognosis in the associated cardiovascular morbidities such as heart failure (HF) [33] and aortic stenosis [34]. The 2% VAF threshold for defining CHIP was originally based on the technical sensitivity of exome sequencing [28,35]. However, advances in sequencing technology now allow for the detection of a VAF as low as 0.01% in mutations [36], providing more detailed clinical associations [36,37,38,39].

### 2.3. Molecular Insights and Pathophysiology

The most frequently mutated gene in CHIP is DNA methyltransferase 3a (DNMT3A). DNMT3A codes for a methyltransferase enzyme that catalyses DNA methylation at CpG sites and plays a crucial role in the regulation of epigenetics. The pathogenic mutations typically result in loss of function, including disruptive missense mutations, nonsense mutations, insertions–deletions, and splice site mutations in regulatory and catalytic domains. These mutations promote the self-renewal of haematopoietic stem cells [40] and enhance multipotency gene expression while subduing differentiation factors [41]. DNMT3A mutations can impact all haematopoietic lineages, inducing proinflammatory T-cell polarisation and activating the inflammasome complex.

TET2 CHIP carriers were found to have significantly increased serum IL-1β levels in an in silico analysis of the TOPMed group [42]. The cardiovascular risk presented by DNMT3A- and TET2-mutant CHIP can be mitigated by an inhibitory IL-6 receptor gene variant (IL6R p.Asp358Ala) [43], highlighting the key role of the mediators of inflammation NLRP3, IL-1β and IL6 in CHIP-associated atherosclerosis. TET2 CHIP carrier status was associated with increased circulating levels of non-classical monocytes (CD14dimCD16^++^), which secrete higher concentrations of proinflammatory cytokines such as TNF-α, IL-1β, and IL-8 in patients with severe degenerative aortic stenosis undergoing transcatheter aortic valve implantation (TAVI). These patients exhibited elevated medium-term overall mortality following TAVI [34]. An association between TET2 CHIP carrier status, HF progression, and worse clinical outcomes was demonstrated through deep-targeted amplicon sequencing of bone-marrow-derived mononuclear cells in patients with chronic HF, with a dose–response relationship observed with increasing TET2 VAF [33].

JAK2 p.V617F mutations in CHIP tend to occur at a younger age and are associated with a 10-times elevated risk of CAD, representing the highest risk of premature heart disease among the CHIP variants [27,42]. PPM1D loss-of-function mutations are particularly associated with clonal haematopoiesis in patients who were previously treated with cytotoxic therapies such as cisplatin, etoposide, and doxorubicin [44]. p53 mutations stimulate haematopoietic stem cell (HSC) expansion under radiation-induced stress, and the p53 mutant interacts epigenetically with EZH2 to promote chromatin association, increasing the H3K27 tri-methylation of the genes responsible for regulating the differentiation and self-renewal of HSCs [45].

### 2.4. CHIP in Malignancy and Therapeutic Implications

Clonal haematopoiesis is frequently observed in various cancers, often as a result of cytotoxic therapies. The incidence of clonal haematopoiesis was 30% among patients with non-Hodgkin’s lymphoma treated with stem cell transplantation [46]. Cancer treatments including ionising radiation, topoisomerase II inhibitors, and cisplatin select for mutations in DNA damage response genes TP53, PPM1D, and CHEK2 [44,47]. Notably, immune checkpoint blockade does not appear to be associated with the expansion of clonal haematopoiesis [48]. A group of 135 patients with invasive glioma receiving temozolomide treatment underwent the next-generation sequencing of cfDNA, and an elevated incidence of CHIP-type mutations was observed. The most frequent mutations occurred in TP53, followed by ATM, GNAS, and JAK2. These mutations were associated with shorter survival [49]. Therefore, assessing the cardiovascular risk associated with increased CHIP prevalence in cancer survivors is essential for optimising long-term oncologic care and survivorship outcomes.

### 2.5. Drivers and Implications of Clonal Haematopoiesis

There are unique conditions that driver clonal haematopoiesis. Whilst the rate of mutation per DNA replication stays the same [50], HSCs replicate with ageing; by 70 years of age, there is a projected accumulation of up to 1.4 million coding mutations within the HSC pool [51]. Other conditions that stimulate clonal haematopoiesis include reactive oxygen species [52], smoking [53], and chemotherapy [47], which induce DNA mutations, chronic inflammation [43,54], chronic infections [55], HIV [56], certain germline mutations [42,57], and atherosclerosis [58], which chronically drive HSC clone formation. Radiation therapy, topoisomerase II inhibitors (e.g., anthracyclines), or platinum therapeutics select clones with mutations in PPM1D and p53. These mutations confer a survival advantage to the mutated HSCs, allowing them to outcompete non-mutated HSCs [47,59,60].

Dyslipidaemia, traditionally a cardiometabolic consequence in obesity, may also be observed in patients without obesity within the atherosclerosis trait complex [58]. When stimulated with low-density lipoprotein, *Tet2*-deficient macrophages in atherosclerotic plaque produce more IL-1 and IL-6 compared to macrophages without *Tet2* mutations [27,61], suggesting that hypercholesterolaemia may intensify the proinflammatory effects of CHIP. Additionally, clonal proliferation can be stimulated by low-circulating high-density lipoprotein [62,63], high-intracellular cholesterol in HSCs [58], and atherosclerosis itself [58,64]. Smoking induces DNA mutations and increases haematopoietic proliferation [65], making it a possible cause of CHIP, especially for clonal haematopoiesis driver mutations (CHDMs) in ASXL1 [47,66].

The detection of specific CHIP mutations may improve the precision of hypercholesterolaemia therapy prescription, particularly for the more costly therapies such as proprotein convertase subtilisin/kexin type 9 (PCSK9) inhibitors. Directing these therapies to patients with detected CHIP mutations with the aim of aggressive lipid control may lead to better cardiac outcomes and improved cost effectiveness. When using guidelines to optimise the lipid profile, considering CHIP as an additional “risk-enhancing factor” influencing statin initiation has been suggested [4,67]. The cardioprotective potential of CHIP-guided statin initiation requires assessment through randomised clinical trials incorporating clinically relevant cardiovascular endpoints. 

CHIP is intricately linked with the development and progression of the cardiovascular risk factors that stimulate clonal proliferation and CHIP, as detailed above. Cardiovascular risk optimisation is crucial, and CHIP can potentially be used for stratifying risk and guiding therapy. Independent of cardiovascular risk factors, patients with CHIP have doubled risk for CAD and stroke [28] and up to quadrupled risk for early-onset (<50 years of age) myocardial infarction [27]. Patients with established ASCVD also have an increased propensity for atherosclerotic events when they have CHIP [68]. This highlights CHIP as a prominent novel risk factor for ASCVD, with up to 17% of people with CAD having clonal haematopoiesis [27,69]. Three processes are crucial in the interface between CHIP and ASCVD: (1) CHIP upregulates the inflammasome/IL-1/IL-6 pathway; (2) CHIP increases ASCVD risk in a dose-dependent manner, with larger mutated leukocyte clones being associated with higher ASCVD risk; (3) atherosclerosis promotes CHIP progression.

Emerging evidence suggests that CHIP contributes to the development and prognosis of HF, both in ischaemic and non-ischaemic contexts [34,37,38,39,70,71,72,73,74,75,76,77,78,79]. CHIP-related mutations can drive this process through three primary mechanisms: (1) disrupting the equilibrium between HSC self-renewal and differentiation [40,80], (2) increasing HSC resistance to external stressors (e.g., chemotherapy) [47,59,60], and (3) providing resistance to inflammation [54]. Each CHDM may have a distinct process that promotes the dominance of HSCs. These mutations are most frequently isolated in genes that encode for epigenetic enzymes (e.g., DNMT3A, TET2, and ASXL1), signalling proteins (e.g., JAK2) [28,29,81], spliceosome components (e.g., SRSF2 and SF3B1), or components of the DNA damage response (e.g., PPM1D and TP53) [28,29,81].

Clonal haematopoiesis is associated with mortality in patients with heart failure with a reduced ejection fraction (HFrEF), independent of age [38,39,78]. Non-ischaemic HFrEF is a varied group of cardiomyopathies that includes dilated cardiomyopathy and hypokinetic nondilated cardiomyopathy, excluding CAD as a cause. CHIP predicts the incidence of HF, mainly in patients without previous ASCVD [82]. CHIP was found to elevate the risk of ensuing development of HF by 25% [82]. Although information on coronary arteries and LVEF status at the time of HF onset is lacking, CHIP was suggested to predict the subsequent onset of non-ischaemic HFrEF, especially since JAK2V617F mutations are associated with a reduced LVEF [82]. Several of the mechanisms present in non-ischaemic HFrEF are anticipated to be analogous to the mechanisms of CHIP in ischaemic HFrEF. CHIP could represent a new target for therapy in non-ischaemic HFrEF. Targeting the immunological response through the use of anti-inflammatory drugs and other immunotherapies could introduce a novel dimension to HFrEF therapy [83].

Heart failure with a preserved ejection fraction (HFpEF) is a vastly multifaceted, multiorgan entity with varied pathophysiological phenotypes [84,85]. Inflammation is recognised as a key disease driver and a promising focus for therapy [86,87,88,89,90,91,92,93,94,95]. However, HFpEF encompasses multiple patient phenogroups, and some of which are not characterised by an increase in inflammation. CHIP may be relevant in some of the HFpEF phenogroups. A recent prospective study in a population-based cohort in Groningen, The Netherlands, demonstrated that CHIP predicted the occurrence of HFpEF in patients under 65 years of age [79]. Those with CHIP had double the risk of developing HFpEF compared to those without, while CHIP was not associated with an elevated risk of HFrEF [79]. Despite the ageing-related nature of CHIP and its association with various comorbidities, CHIP may serve as an independent risk factor for HFpEF development in individuals under 65 years of age [79]. In addition, CHIP forecasted the onset of HF, but common CHDMs, such as DNMT3A and TET2, did not forecast LVEF reduction, suggesting that CHIP predicts HFpEF onset in a subset of patients with CHIP [82].

### 2.6. Use Case 3: Polygenic Risk Scoring

GWASs have identified over 250 genetic variants linked to CAD, advancing our understanding of its pathogenesis [96]. These disease-associated variants can be aggregated into PRS, which are innovative genomic tools that enhance disease prediction, offering the unique benefit of being computable in early life. Recent advances in computational methods have led to the development of expanded (“genome-wide”) polygenic scores (GPSs), which incorporate genetic variants in the millions. These comprehensive scores provide a more accurate assessment of extreme CAD risk and may offer a risk axis independent of traditional CAD risk factors [97,98].

The unique utility of a genetic predictor lies in its ability to quantify disease risk from birth and across an extended time horizon, prior to clinical risk factors emerging. This suggests that CAD PRS may have the greatest utility in younger individuals. When combined with cardiotoxic therapy, a corresponding genetic predisposition can substantially elevate the risk of cancer-therapy-related cardiovascular disease. Given that genetic markers, particularly polygenic scores, are now available for several cardiovascular conditions, aligning cancer-therapy-induced complications with appropriate genetic predictors for these conditions represents a promising approach. Conducting genetic testing once allows for the calculation of multiple polygenic scores, thereby optimising cost-effectiveness and increasing the overall utility and adoption of clinical genetic testing to improve outcomes for cancer survivors.

## 3. Benefits and Impact on Patient Care

### 3.1. Cancer-Therapy-Related Cardiac Dysfunction

A meta-analysis conducted by Lee et al. [99] examined 35 studies, encompassing a total of 219 single-nucleotide polymorphisms (SNPs) across 80 genes, 11 antineoplastic agents from five drug classes—including tyrosine kinase inhibitors, monoclonal antibodies, antimetabolites, alkylating agents, and immunomodulatory agents—and five cardiovascular toxicities of reduced LVEF, hypertension, cardiac arrhythmia, venous thromboembolism, and cardiovascular disease. A total of 34% of the SNPs in 40 genes were found to be significantly associated with a risk of particular CTRCDs [99]. Notably, the rs1136201 and rs1058808 polymorphisms of HER2 were identified as possible predictors of trastuzumab-related cardiotoxicity.

These findings hold significant clinical relevance as they enable predictions that can facilitate the development of individualised treatments for specific patient populations or groups of individuals. The advent of bespoke therapy tailored to the individual patient is now feasible, particularly with options such as cardiac-sparing chemotherapy regimens, alternative anthracycline analogues such as liposomal doxorubicin, and cardioprotective therapies like carvedilol, dexrazoxane, or enalapril [100,101]. Given the prevalence of cardiovascular toxicity among patients receiving cancer treatments, it is plausible to employ advanced approaches such as human GWASs or whole-exome or whole-genome sequencing, accompanied by appropriate population stratification, to direct established cardioprotective strategies to individuals identified as high risk for CTRCD [102].

Recent network-based approaches offer rapid means to identify drugs and understand toxicities [103]. For example, genetic interactions (gene–gene co-mutations) were shown to be more precise markers for predicting sensitivity or resistance to oncological treatments compared to using mutant genes alone in a pan-cancer genetic network analysis using 6700 whole-exome sequencing data from The Cancer Genome Atlas [104]. Integrating large-scale patient genetic, genomics, transcriptomics, and proteomics data with cutting-edge network medicine methodologies has the potential to drive significant advancements in cancer pharmacogenomics and precision cardio-oncology [105,106].

### 3.2. Clonal Haematopoiesis of Indeterminate Potential

As the genotypic and phenotypic associations with CHIP are unravelled, researchers and clinicians can then in turn consider preventative and therapeutic strategies to moderate disease risk. A precision-medicine-based approach to risk reduction can be adopted by understanding which patient cohort would be appropriate for targeted CHIP testing. Moreover, many patients with CHIP are often identified incidentally. As the majority of cell-free DNA originates from haematopoietic cells, CHIP frequently confounds the identification of circulating tumour DNA during cell-free DNA analysis in early cancer detection [107,108,109].

The understanding of CHIP biology and its connection with a proinflammatory state has unveiled several promising therapeutic targets. Notably, the presence of the inhibitory IL-6 receptor gene variant (IL6R p.Asp358Ala) has been shown to reduce CVD risk in DNMT3A and TET2 CHIP carriers by ~50%, suggesting that the inhibition of IL-6 signalling can significantly reduce CAD risk among individuals with CHIP compared to those without [43]. This finding is consistent with the Canakinumab Anti-inflammatory Thrombosis Outcomes Study (CANTOS), demonstrating that IL-1β blockade with canakinumab after myocardial infarction (MI) reduced the risk of death from CVD, as well as rates of non-fatal AMI and non-fatal stroke. The data from CANTOS suggest that canakinumab, a monoclonal antibody targeting IL-1β, may prevent major adverse cardiovascular events (MACEs) in patients with prior MI and elevated C-reactive protein [69,110]. Further exploratory analyses revealed that patients with *TET2* mutations had a lower risk of MACEs while on canakinumab compared with a placebo [69], while post hoc analyses suggested an even greater reduction in adverse events in those with TET2 CHIP [111]. In a JAK2 CHIP model, both non-selective IL-1 receptor blockade with anakinra and targeted IL-1β blockade attenuated the instability of atherosclerotic plaque, normalised the macrophage proliferation and density in early atherosclerotic plaques, and attenuated plaque core necrosis while increasing the thickness of the plaque cap in advanced atherosclerotic plaques [112]. It has been suggested that there is a dose–response relationship between VAF (a marker of clone size, where a VAF of 1% corresponds to mutations in 2% of leukocytes) and ASCVD [27,28,43,69]. Patients with a higher VAF had a higher coronary artery calcium score [27] and an increased risk of MACEs [28,43]. This dose-dependent effect of the VAF was also observed in patients with a genetic *IL6R* deficiency and *DNMT3A* or *TET2* CHDMs [43], with the effect primarily seen in patients with larger clone sizes. Thus, when selecting patients for anti-inflammatory therapy, patients with a higher VAF might be better candidates. Human studies have shown that atherosclerosis increases the markers of proliferation in HSCs, while cholesterol levels remain normal, suggesting increased proliferation and acceleration of CH [58]. The inflammasome/IL-1/IL-6 pathway is pivotal in CHIP-induced ASCVD, making anti-inflammatory treatments potentially significant in reducing cardiovascular risk in patients with CH.

The tumour necrosis factor family is upregulated in a subgroup of patients with HFpEF with multiple comorbidities such as obesity and diabetes [94,113]. TNF-α stimulates clonal expansion with myeloid skewing, at least in vitro [114], and increased TNF-α levels were noted in the circulating monocytes of patients with heart failure (including those with aortic stenosis) and *DNTM3A* mutations as well as in pressure-overload mice models with *Jak2* mutations [73,115]. Previous TNF-α trials did not identify improved outcomes in patients with HFrEF [116,117]. Targeted CHIP-based patient selection could assist in delineating a patient subgroup for who this treatment may prove more effective.

Targeted inhibition of the inflammasome may also protect against atherogenesis. Pharmacologic inhibition of the NLRP3 inflammasome with MCC950 resulted in a 50% reduction in the size of the aortic atherosclerotic plaque in TET2-deficient mouse models. This effect was larger than the non-significant reduction seen in wild-type mice [61]. Similarly, targeting the AIM2 inflammasome, characteristic of JAK2-mutant CHIP, might offer comparable benefits [112]. Driver mutations in CHIP could present a different therapeutic target. Vitamin C metabolites stimulate TET2, mimicking the restoration of TET2 by promoting 5-hydroxymethycytosine formation in TET2-deficient mice, potentially serving as a prevention therapy for carriers of TET2-CHIP [80]. In JAK2-mutant CHIP, treatment with the JAK2 inhibitor ruxolitinib reduced the formation of abnormal neutrophil extracellular traps and deep vein thrombosis [118] as well as the levels of IL-18 in the circulation [112]. JAK2 inhibition with fedratinib in Apoe^−/−^ mice attenuated myelopoiesis and atherosclerotic development [119].

A key direction for future research is understanding the relationships of race, ethnicity, and ancestry with the prevalence of CHIP. Most initial CHIP research has focused on European Caucasian groups. CHIP prevalence has been observed to be relatively lower, however, in people of Hispanic and East Asian ancestry when compared with people of other ancestries [27,43]. The germline variant rs144418061, which increases CHIP risk, is found only in people of African ancestry [43], recognised partly due to the wider diversity of the TOPMED group (40% European, 32% African, 16% Hispanic, and 10% Asian). Future research in cohorts that are racially and ethnically diverse will be essential to increase our understanding of the consequences of a CHIP diagnosis and may assist in identifying additional relationships between specific ethnic groups and germline variants or candidate driver mutations.

### 3.3. Polygenic Risk Scoring

Polygenic susceptibility to CAD appears to be substantially attenuated by the adoption of healthy lifestyle behaviours. This underscores the benefit of healthy lifestyle measures—such as avoidance of smoking, optimal weight control, sufficient exercise, and balanced dietary choices—among survivors of childhood cancer who received prior treatment with radiation therapies [120]. In addition, several post hoc clinical trial analyses have shown the significant benefit of lipid-lowering therapies, including statins and PCSK9 inhibitors, for individuals with an increased polygenic CAD risk [121,122].

When these findings are extrapolated to the subgroup identified by Sapkota et al. [123], earlier statin therapy may be particularly beneficial for this high-risk population. Currently, the suitability of polygenic risk prediction for CAD as a screening test for the general population is debatable. However, the present study suggests the existence of distinct subgroups and clinical scenarios where such polygenic risk assessments could be valuable. Additionally, the novel application by Sapkota et al. [123] of a GPS to young, high-risk survivors of cancer provides a framework for advancing preventive cardio-oncology.

### 3.4. Long Noncoding RNAs and microRNAs as Crucial Regulators in Cardio-Oncology

Circulating micro-RNAs, products of specific genomic profiles, show considerable potential for identifying subclinical cardiotoxicity in patients receiving certain therapies. These noncoding small RNAs circulate in the bloodstream, enter distant recipient cells, and regulate gene expression, demonstrating they could be biomarkers of cardiovascular disease.

In a case–control study involving 12 children, 84 microRNAs were profiled 24–48 h and approximately 1 year after the initiation of anthracycline chemotherapy. The study identified an association between a reduced LVEF and three specific microRNAs [124]. Another study involving 33 children receiving either anthracycline chemotherapy or non-cardiotoxic chemotherapy profiled 24 micro-RNAs at pre- and post-cycle time points. The findings showed greater chemotherapy-induced dysregulation in patients receiving anthracyclines compared to those receiving non-cardiotoxic chemotherapy [125]. These studies highlight the critical role of microRNAs in monitoring and potentially mitigating cardiotoxic effects in paediatric oncology patients.

Noncoding RNAs (ncRNAs), comprising long noncoding RNAs (lncRNAs) and microRNAs (miRNAs), play a pivotal role in the interplay between anticancer drugs and cardiovascular complications. ncRNAs have the potential to serve as novel biomarkers for cancer-drug-induced cardiotoxicity or as therapeutic targets to mitigate such effects. lncRNAs are integral to the progression of both cancer and cardiovascular disorders, influencing gene and protein expression across various cellular signalling pathways. A key mechanism through which lncRNAs contribute to the pathologies of cancer and cardiovascular disease is mitochondrial dysfunction. Some lncRNAs offer protection against cancer-therapy-induced cardiotoxicity, while others are linked to an increased risk of such adverse effects. Despite their presence in these diseases, our understanding of lncRNAs remains limited.

In cancer, miRNAs can function as either oncogenes or tumour suppressors. Oncogenic miRNAs target and downregulate tumour suppressors, promoting cell proliferation and tumour growth. Conversely, other miRNAs interfere with cellular proteins to inhibit oncogenic activity. Similar to lncRNAs, certain miRNAs can either protect against or increase the risk of cancer-therapy-induced cardiotoxicity. Cancer treatments are known to modulate miRNA levels, and the differential expression of specific miRNAs may influence the occurrence of cardiotoxicity. Circulating ncRNAs show potential for the development of new diagnostic and prognostic markers of cardiovascular disease. lncRNAs, in particular, show promise as diagnostic tools for predicting anticancer-therapy-induced cardiotoxicity. Differentially regulated ncRNAs, especially lncRNAs, could become new therapeutic targets to counteract the cardiotoxic effects of anticancer drugs, ideally without compromising the efficacy of the treatment or disease progression. The integration of precision genomics into cardio-oncology clinical care is essential for the advancement of this emerging field.

### 3.5. Modelling of Precision Cardio-Oncology Through the Use of Human-Induced Pluripotent Stem Cells (hiPSCs) for Risk Stratification and Prevention

hiPSCs offer an unparalleled model for the in vitro testing of human genetics. These cells are created by reprogramming differentiated adult cells using a combination of genes that revert them to an embryonic-like, undifferentiated state [126]. Once reprogrammed, hiPSCs can be differentiated into various cell types, such as cardiomyocytes, endothelial cells, and fibroblasts, through the use of small molecules and growth factors. This process allows for the generation of an unlimited number of cardiomyocytes from a patient’s cells in the laboratory, enabling personalised therapeutic testing without the need for cardiac biopsies to harvest cardiomyocytes.

hiPSC-derived cardiomyocytes have been instrumental in characterising cellular disease phenotypes [127], facilitating high-throughput drug discovery [128], and investigating cardiovascular metabolism [129], among many other functions [130], thus cementing their key role in contemporary cardiovascular research.

Furthermore, the combination of hiPSCs with genome editing technologies such as CRISPR-Cas9 enables the creation and study of various genetic variants in the laboratory. By introducing disease-causing mutations into the genomes of healthy hiPSCs and then differentiating them into cardiomyocytes, researchers can investigate whether these mutations result in disease phenotypes. Conversely, gene editing can be used to correct suspected disease-associated mutations in patient-derived hiPSCs to determine if the correction alleviates disease phenotypes. These strategies provide additional evidence to validate the presumptive mutations and highlight them as possible targets in affected patients [131].

A landmark study demonstrating the therapeutic potential of CRISPR editing in hiPSC corrected the disease-associated mutations in hiPSCs derived from patients with β-thalassemia [132]. In this condition, mutations in the human haemoglobin beta (HBB) gene lead to severe anaemia due to reduced β-globin production. CRISPR-corrected hiPSCs, when differentiated into erythroblasts, exhibited normal levels of HBB expression and could serve as a source for autologous transplantation [132]. Although this therapeutic approach carries risks, such as the possibility of residual undifferentiated cells forming tumours, it showcases the immense potential to correct patient-specific mutations in their own cells.

Further study of novel therapies in cardio-oncology using hiPSC models, including CRISPR editing strategies, holds significant promise. By leveraging hiPSCs to create personalised models and evaluate the cardiotoxicity of individual treatments, we can develop more patient-specific treatment plans in precision cardio-oncology, ultimately optimising clinical outcomes.

## 4. Challenges and Mitigation Strategies

### 4.1. Ethical, Legal, and Societal Implications

The ethical, legal, and societal implications of genetic testing, as well as the use of genetic information in precision medicine, must always be duly considered. The underlying principle in considering these issues is balancing the benefits against the potential harm to patients, their families, and society at large within the framework of national regulations and ethical standards. This includes examples such as genetic testing in phenotype-negative individuals, genome sequencing in otherwise healthy individuals, and gene editing in humans. While these advancements have resulted in significant progress in genetic technologies, they have also raised substantial ethical, legal, and societal concerns. Addressing these issues requires the availability of multidisciplinary expertise to comprehensively serve the needs of patients, their families, and the community. Furthermore, facilitating open and informed discussions between patients and clinicians about the implications of rapidly evolving genetic technologies is essential.

### 4.2. Integration of Precision Genomics into Clinical Practice

One common challenge in the clinical incorporation of precision genomics is prioritising its integration within the health system’s electronic health records (EHRs). A key mitigation strategy involves employing data warehousing techniques to implement specific genomics tools within EHR systems. By extracting data from multiple sources and integrating clinical records into a central repository, hospitals can adapt and tailor genomic innovations to various contexts [133]. This approach not only enhances the integration of genomics into healthcare but also ensures that the benefits of these advancements can be realised across different healthcare settings.

### 4.3. Engaging Patients in Genomic Medicine Projects

Engaging patients effectively in genomic medicine projects presents another challenge. To overcome this, various strategies can be implemented. Mass media can be utilised to communicate the benefits and innovations of genomic medicine to a wide audience, thereby increasing public awareness [133]. Active patient involvement in implementation activities, such as through a patient advisory board, can also be beneficial. Moreover, personalised strategies to enhance patient engagement can prepare them to be actively engaged in their healthcare decisions. This involves training patients to ask informed questions about the evidence behind clinical decisions, thus empowering them to take an active role in their care [133].

### 4.4. Funding and Staffing for Precision Genomic Services

The establishment and maintenance of precision genomic services require significant funding and skilled personnel. Securing funding support from hospital leadership is essential to cover the setup and operational costs. Additionally, developing a pipeline for future talent is crucial for sustaining these services. In this regard, the M.S. in Precision Medicine course at the National University of Singapore has been established to provide a pipeline for future talent, and the proposed cardio-oncology service should establish working relationships with students and faculty through internship programmes that nurture geneticists for future onboarding.

### 4.5. Addressing Clinical Implementation and Access

When new clinical genetic or genomic applications are validated, several issues must be addressed to ensure proper clinical implementation and utilisation. Ensuring that patients have timely access to newly available or FDA-approved tests is critical to avoid diagnostic delays and suboptimal care [134]. Medical care delivery involves multiple stakeholders, and changes in practice behaviour and patient acceptance can be slow. Recognising that the translation of new applications into practice often takes longer than the development and validation phases is important for setting realistic expectations and planning accordingly. By addressing these challenges with comprehensive strategies, the integration of genomic medicine into clinical practice can be effectively facilitated, benefiting patients, clinicians, and the broader healthcare system.

### 4.6. Need for Diverse Study Populations

Clinical differences in disease onset, severity, outcomes, and drug response may be due to population-specific variables such as diet, occupation, genetic variant prevalence, and social determinants of health. Therefore, varied study groups are essential for obtaining a comprehensive understanding and creating clinically useful applications, especially in heterogeneous populations. However, as also seen in many clinical studies [135], a large majority of study subjects in genetics and genomics research are of European ancestry [136,137]. This paucity of information from under-represented groups regarding allelic prevalence and association strength with specific phenotypes (e.g., disease risk, drug response) limits the use of genetic and genomic applications to certain groups due to the uncertain positive predictive value of these tests [138].

### 4.7. Regulatory Oversight of Genetic and Genomic Technologies

The regulatory overview of genetic and genomic testing is complicated and often lags behind the rapid advancements in technology and analysis [139]. This complexity arises from the type of test, its manufacturing and sale processes, and the discretionary enforcement policies in place. With the advance of liquid biopsies and sequencing platforms, the regulatory framework continues to lag [140].

Information on an individual’s genetic disease risk could be exploited by employers, schools, insurers, or other groups, raising significant concerns. This issue was first highlighted shortly after the inception of the Human Genome Project [141,142], although workplace discrimination based on genetic information predates the project. Despite the existence of legal protections, public awareness of these protections remains low [143,144,145,146,147,148]. Concerns about genetic discrimination also vary depending on the type of testing and the patient population [149]. Internationally, genetic discrimination is still a key matter of interest, even in countries with national protective frameworks. Furthermore, many U.S. healthcare providers do not have a thorough comprehension of the existing protections (or lack of protection) regarding genetic information, often failing to communicate this information to patients [150,151,152]. Discussing the risk of discrimination is recommended for patients considering genetic testing [153,154].

### 4.8. Enhancing Clinicians’ Knowledge and Beliefs

Provider education is crucial for the appropriate use of any new clinical innovation, including genetic and genomic applications [155]. While healthcare providers have shown interest in these new applications, studies have consistently reported a limited knowledge base for their appropriate use [156,157,158,159,160,161,162]. This knowledge deficit spans various healthcare professionals, including pharmacists, physicians, and nurses, which is not surprising given the recent shift toward personalised medicine and the limited teaching of the topic in medical school curricula. As a result, healthcare providers may be hesitant to use emerging technologies if they are underprepared to discuss them with patients or are unsure about how to integrate new information into treatment decisions. The extensive roll-out of electronic medical records offers point-of-care clinical decision support (CDS), providing timely data about the genetic and genomic tools for guiding therapeutic strategies [163,164,165,166,167]. CDS can support traditional continuing education methods and provide rapid, patient-specific information. However, changing provider behaviours may require additional support due to the potential for alert fatigue, which can diminish the value of CDS. A favourable attitude towards a CDS system for pharmacogenomic testing is associated with a stronger desire to upskill and to implement emerging genetic/genomic tests in clinical practice [157].

## 5. Conclusions and Future Directions

Advancements in precision cardio-oncology, particularly in genetic testing and therapeutics, enable clinicians to better assess risk, diagnose conditions, and deliver personalised, cost-effective therapeutics. Through case studies of CTRCD, CHIP, and PRS, we demonstrated the benefits of incorporating precision genomics in individualised care in cardio-oncology. Furthermore, leveraging real-world genomic data in clinical settings can advance our understanding of long noncoding RNAs and microRNAs, which play key regulatory roles in cardio-oncology. Additionally, employing hiPSCs for risk stratification and prevention represents a promising avenue for the modelling of precision cardio-oncology. While these advancements have led to significant progress in genetic approaches, they have also raised substantial ethical, legal, and societal concerns. Regulatory oversight of genetic and genomic technologies should therefore evolve suitably to keep up with the rapid advancements in technology and analysis. Provider education is crucial for the appropriate use of new genetic and genomic applications, including on the existing protections available for patients regarding genetic information. This can provide the confidence required for diverse study groups to step forward for genetic studies looking to obtain a comprehensive understanding and determine effective clinical applications for heterogeneous populations. In clinical settings, the implementation of genetic and genomic applications within electronic medical records can offer point-of-care clinical decision support, thus providing timely information to guide clinical management decisions.

## Figures and Tables

**Figure 1 ijms-26-02052-f001:**
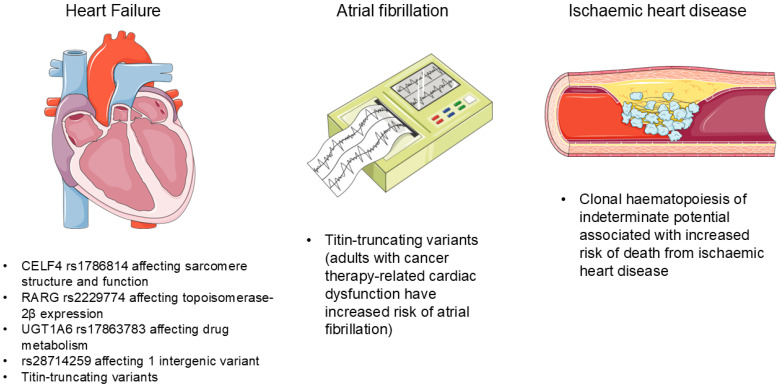
Key genetic variants and associated cancer therapy-related cardiovascular complications.

**Table 1 ijms-26-02052-t001:** Genetic considerations in baseline cardiovascular toxicity risk assessment before cancer therapy initiation.

Cancer Therapy-Related Cardiovascular Complication	Cardiovascular Assessment	Genetic Rationale for Cardiovascular Assessment
Cancer therapy-related cardiovascular toxicity (CTR-CVT)	Family history of premature cardiovascular disease (CVD)	Genetic abnormalities associated with CVD may predispose patients with cancer to a higher risk of CTR-CVT.
QT prolongation due to cancer treatment	Baseline electrocardiogram (ECG)	When baseline QT prolongation is identified, the correction of reversible causes and the identification of genetic conditions that prolong QT is recommended.
Drug-induced QT prolongation and Torsades de Pointes (ventricular arrhythmia)	Family history of sudden cardiac death (SCD) (congenital long QT syndrome (LQTS) or genetic polymorphism)	Family history of SCD (congenital LQTS or genetic polymorphism) is a risk factor for drug-induced QT prolongation and Torsades de Pointes (ventricular arrhythmia).
Atrial fibrillation (AF) associated with cancer	Genetic predisposition to AF	Genetic predisposition to AF contributes to pathophysiology of AF associated with cancer.
Cancer-associated thrombosis	Genetic predisposition of patient to thrombosis and genetic characteristics of cancer (JAK2 or K-ras mutations) that increase thrombosis risk	Risk of cancer-associated thrombosis is influenced by genetic predisposition of patient to thrombosis and genetic characteristics of cancer (JAK2 or K-ras mutations) that increase thrombosis risk.

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
