# Peer review of "Revolutionising Cardio-Oncology Care with Precision Genomics"

_ijms, 2025, doi:10.3390/ijms26052052_

Round 1
Reviewer 1 Report
Comments and Suggestions for Authors
Thank you for this work. As GWS testing becomes more accessible and used in practice, the ability to use genetic information to support cardiac care will be crucial to address the side effects of chemotherapy and other medications.
The review outlines some areas where genetics may play a role and some of the early research in this space. The work overstates some of the conclusions that can be made as much of the current work shows association rather than causation and the concept that therapy could play a role is still in the realm of research.
The work, given the title, also focuses on the findings related to cardiac care, and while some arguments could be made regarding changing chemotherapy, until clinical trials are done the cardiac benefit vs the change in oncologic care cannot be determined. This needs to be included in the paper.
While the work related to cardiac genetics is of benefit, the later parts of the paper related to challenges and mitigation strategies (section 4) are superficial. While these points are valuable, the superficial inclusion in this paper does not add anything. These concepts can be briefly addressed, and the paper focus on the current work related to cardio-genetics.
The work comments on arrhythmias and genetics. Atrial fibrillation associated with cancer is specifically called out in table 1, including the statement "genetic predisposition to AF contributes to pathophysiology of AF associated with cancer," but the word fibrillation does not appear in the text of the manuscript. If AF is truly a clinical concern which should be addressed and monitored differently for those with a genetic predisposition, this should be stated within the text of the manuscript.
Author Response
Comment 1: As GWS testing becomes more accessible and used in practice, the ability to use genetic information to support cardiac care will be crucial to address the side effects of chemotherapy and other medications. The review outlines some areas where genetics may play a role and some of the early research in this space. The work overstates some of the conclusions that can be made as much of the current work shows association rather than causation and the concept that therapy could play a role is still in the realm of research. The work, given the title, also focuses on the findings related to cardiac care, and while some arguments could be made regarding changing chemotherapy, until clinical trials are done the cardiac benefit vs the change in oncologic care cannot be determined. This needs to be included in the paper.
Response 1: We thank Reviewer 1 for this insight. We have modified the manuscript to reflect the need for clinical trials to interrogate the efficacy of precision medicine approaches in improving cancer and cardiac outcomes. We have modified the manuscript to reflect the need for randomised clinical trials incorporating clinically-relevant cardiovascular endpoints to assess the cardioprotective potential of CHIP-guided statin initiation. Kindly please see tracked changes and associated comments in revised manuscript. The options of cardiac-sparing chemotherapy regimens, alternative anthracycline analogs such as liposomal doxorubicin, and cardioprotective therapies like carvedilol, dexrazoxane, or enalapril are already established clinical strategies to mitigate cancer therapy-related cardiac dysfunction. It is plausible to employ gene sequencing approaches to direct these established cardioprotective therapies to high-risk individuals.
Comment 2: While the work related to cardiac genetics is of benefit, the later parts of the paper related to challenges and mitigation strategies (section 4) are superficial. While these points are valuable, the superficial inclusion in this paper does not add anything. These concepts can be briefly addressed, and the paper focus on the current work related to cardio-genetics.
Response 2: We thank Reviewer 1 for this comment. As a reflection of the clinical backgrounds of the authors and to cater to clinical genomics queries of potential clinically-inclined readership, we have included this section addressing key points on ethical/legal implications, integration and implementation of genomics into clinical practice, engaging patients and staff, diversity of study populations, regulatory oversight as well as continuing genomic education among clinicians. We believe that this section may be of some use and interest to clinicians who may read this review. Kindly please see associated comment in revised manuscript.
Comment 3: The work comments on arrhythmias and genetics. Atrial fibrillation associated with cancer is specifically called out in table 1, including the statement "genetic predisposition to AF contributes to pathophysiology of AF associated with cancer," but the word fibrillation does not appear in the text of the manuscript. If AF is truly a clinical concern which should be addressed and monitored differently for those with a genetic predisposition, this should be stated within the text of the manuscript.
Response 3: We thank Reviewer 1 for this suggestion. In response to Reviewer 1’s insights (comment 3), we have stated that clinicians should have a higher index of suspicion for the development of AF in cancer patients with associated genetic predisposition. Kindly please see associated comment in revised manuscript.
Reviewer 2 Report
Comments and Suggestions for Authors A very large and complex work, in truth it is more of a research project than a clinical or laboratory study, so it is not easy to place in specialist journals. A big problem we have today is that in the face of thousands of genetic data related to oncological diseases and we do not know exactly how these alterations are useful in choosing the best therapies because not all the genes found are active or actionable. what is completely missing in this article, I would say research project, is a correlation with common clinical aspects observable every day such as: EKG holter and blood pressure, CPET cardiopulmonary test, lactat test for example and complete spirometry, correlations with ultrasound. And it must be considered that theoretically a patient with many genetic alterations favoring cardiac toxicity could be in perfect cardio-respiratory balance and judged a healthy person. If he has cancer, don't you treat it for a supposed alteration of genes that may not be actionable?Author Response
Comment 1: What is completely missing in this article, I would say research project, is a correlation with common clinical aspects observable every day such as: EKG holter and blood pressure, CPET cardiopulmonary test, lactat test for example and complete spirometry, correlations with ultrasound.
Response 1: We thank Reviewer 2 for this insight. In response, we have stated that common clinical aspects observable everyday such as ECG/Holter/BP/ECHO should be performed. These tests will shed insight into potential development of cardiotoxicities described in further detail in the use cases outlined in the review. Please refer to associated comment and changes in the revised manuscript.
Comment 2: And it must be considered that theoretically a patient with many genetic alterations favoring cardiac toxicity could be in perfect cardio-respiratory balance and judged a healthy person. If he has cancer, don't you treat it for a supposed alteration of genes that may not be actionable?
Response 2: We thank Reviewer 2 for this perspective. Precision methods in cardio-oncology can help risk stratify and guide early detection and monitoring of cancer patients at-risk of developing CTRCD. Identifying cancer patients with genetic predisposition to cardiotoxicity may help strike a balance between preventing cardiotoxicity and managing cancer effectively. The options of cardiac-sparing chemotherapy regimens, alternative anthracycline analogs such as liposomal doxorubicin, and cardioprotective therapies like carvedilol, dexrazoxane, or enalapril are already established clinical strategies to mitigate cancer therapy-related cardiac dysfunction. It is plausible to employ gene sequencing approaches to direct these established therapies to high-risk individuals that may appear phenotypically "well" from a cardiac point of view. In patients with a genetic predisposition, a standard dose of cancer therapy may induce cardiotoxicity, while a reduced dose may compromise cancer outcomes. Therefore, a personalised genetic approach can help determine individual susceptibility to cardiovascular disease in cancer patients, thus allowing clinicians to potentially balance the efficacy and toxicity of cancer drugs.